# Photonic crystals with rainbow colors by centrifugation-assisted assembly of colloidal lignin nanoparticles

Jinrong Liu [1,2], Mathias Nero [1], Kjell Jansson [1], Tom Willhammar[1] & Mika H. Sipponen [1,2] ✉

Photonic crystals are optical materials that are often fabricated by assembly of particles into periodically arranged structures. However, assembly of lignin nanoparticles has been limited due to lacking methods and incomplete understanding of the interparticle forces and packing mechanisms. Here we show a centrifugation-assisted fabrication of photonic crystals with rainbow structural colors emitted from the structure covering the entire visible spectrum. Our results show that centrifugation is crucial for the formation of lignin photonic crystals, because assembly of lignin nanoparticles without centrifugation assistance leads to the formation of stripe patterns rather than photonic crystals. We further prove that the functions of centrifugation are to classify lignin nanoparticles according to their particle size and produce monodispersed particle layers that display gradient colors from red to violet. The different layers of lignin nanoparticles were assembled in a way that created semi-closed packing structures, which gave rise to coherent scattering. The diameter of the lignin nanoparticles in each color layer is smaller than that predicted by a modified Bragg's equation. In situ optical microscope images provided additional evidence on the importance of dynamic rearrangement of lignin nanoparticles during their assembly into semi-closed packing structures. The preparation of lignin nanoparticles combined with the methodology for their classification and assembly pave the way for sustainable photonic crystals.

Shimmering colors have fascinated people throughout the recorded history of humankind[1,2]. Many of the natural colors are structural colors originating from the scattering and diffraction of close-packed particles such as silica in opal[3,4]. More specifically, the periodic structure of a photonic crystal allows specific electromagnetic waves to propagate and specific ones to be forbidden, which gives rise to a photonic band gap[5]. Visible coloration results when the lattice constant of the crystal is comparable to or smaller than the wavelength of light and the bandgap corresponds to a wavelength in the visible range (400–800 nm).

Artificial photonic crystals can be produced by self-assembly of monodisperse particles[6–8]. The assembly process can be manipulated by various external forces, such as gravitational or inertial forces[6], surface tension or capillary forces[9,10], or electric or magnetic fields[11]. A vast majority of these artificial photonic crystals are assembled from colloidal silica[7,12] or polymer spheres (frequently polystyrene or polymethyl methacrylate)[13,14]. However, the aforementioned particles have obvious shortcomings stemming from their sources, production, and end-of-life management. There is thus a growing demand for photonic materials based on renewable biomass.

[1]Department of Materials and Environmental Chemistry, Stockholm University, SE-106 91, Stockholm, Sweden. [2]Department of Materials and Environmental Chemistry, Wallenberg Wood Science Center, Stockholm University, SE-10691, Stockholm, Sweden. ✉e-mail: mika.sipponen@mmk.su.se

Cellulose, hemicellulose, and lignin are the main components of plant biomass. The structural color of cellulose-based optics comes from the helicoidal assembly of cellulose nanocrystals, and there has been significant progress in understanding the key processes involved[15–19]. The development of lignin-based structural colors is still in its early stages[20,21] and further understanding of their formation and properties is highly desired. This advancement is important to pave the way for value-added lignin photonic materials, given that the current practice is to combust lignin in the chemical recovery process of kraft pulp production[22]. Lignin is an aromatic hetero-polymer with a mostly dark-brownish color which originates from its intrinsic chromophores and those formed during the pulping reactions[23]. The brown-black color palette of industrial lignin limits the opportunity window for optical applications[24]. Lignin-retaining bleaching to remove the chromophores is not a fully sustainable strategy due to the dearomatization reactions and generation of degradation products[25].

Structural colors based on lignin could be a way forward as the amphiphilic nature of lignin makes it plausible to form spherical lignin nanoparticles (LNPs) by aggregation in aqueous solvents[26,27]. Solubility-based fractionation can be used to engineer lignin into a value-added material[28–31]. In addition to their renewable origin, one advantage of LNPs is their charge stabilization due to carboxylic and phenolic hydroxyl groups[32], which therefore permits avoiding the use of a surfactant. However, LNPs are not well studied in preparing photonic crystals because of an insufficient understanding of their packing mechanism. In addition, one strict requirement in the preparation of photonics is to have narrow particle size distribution (<5%) for achieving close-packed structures. LNPs do not naturally qualify for this requirement due to their polydispersity. In this sense, Wang et al.[20,21] managed to fabricate monodispersed LNPs by a sol-vent extraction method. Such monodispersed particles could form different color tones with water content 50–75 wt% and show potential applications in smart painting, rewritable paper, and invisible code.

Here we show the possibility of forming lignin photonic crystals with rainbow coloration from non-monodispersed LNPs and improve the understanding of the central processes involved in the formation of the lignin photonic crystals. The rainbow-colored crystals in the size range from 1 mm to 1 cm are produced by centrifugation of LNPs based on commercial soda lignin. Our results show that centrifugation is central to achieving distinct layers of monodispersed LNPs. These results represent a major step towards lignin-based structural colors and tailormade photonic materials.

## Results

### Prediction of particle sizes by Bragg's law

The first question relevant to photonic crystals that we need to answer is which size distribution is proper for lignin-based materials. To answer this question, we can look into the origin of this phenomenon. Structural colors are observed when electromagnetic waves are Bragg-scattered in certain directions owing to a high extent of spatial peri-odicity in the particle lattice[33]. Therefore, the particle size which responds to different visible wavelength scales (at normal incidence) can be predicted from the diffraction of electromagnetic waves fol-lowing a modified Bragg's law[7,34,35]. The calculated particle diameter range that matches the colors from violet to red is from 158 to 311 nm

(Table 1). In other words, when the particle size decreases, the pho-tonic band gap exhibits a blue-shift.

The ideal particle size ($D$) which is theoretically responsible for different colors can be calculated by $D = \frac{\lambda}{2 \times \sqrt{\frac{2}{3}} n_{eff}}$, where $\lambda$ is the wavelength, and $n_{eff}$ is the effective refractive index of lignin. More calculation details are presented in the "Methods" section (calculation of particle size–wavelength relationship).

### Design and preparation of LNPs and photonic crystals

Based on the data in Table 1 we prepared LNPs with a size distribution that we anticipated to be suitable for the formation of structural colors by centrifugation-assisted assembly. The procedure how to prepare LNPs and photonic crystals is shown in Fig. 1. In our case the starting material was soda lignin that was purified by successive solvent extractions first with ethanol and the remaining solids with acetone, following a procedure reported in the literature[20]. The resulting acetone-soluble fraction comprised around 10% of the original soda lignin and was used to prepare LNPs by a solvent-shifting method[36]. The prepared LNPs showed an average hydrodynamic diameter of 184 nm (Fig. 2a) and a polydispersity index (PDI) of 0.19 based on dynamic light scattering (DLS). The theoretical range (158–311 nm) over which LNPs can generate partial photonic band gaps from violet to red is marked by a rainbow color background in Fig. 2a. The hydrodynamic size can be adjusted by varying the water dilution rate (volume of water/volume of lignin solution/total dilution time, i.e., v/v/min) during the solvent shifting method, from 184 to 582 nm, rate from 20 to 0.25 v/v/min (Supplementary Fig. 1). The smallest particles (184 nm) were achieved by the fastest water dilution rate (20 v/v/min), which follows the reported trend in literature[26]. The concentration of the as-prepared dispersion of LNPs was 0.18 wt%. This dispersion was centrifuged to separate small particles which were out of range with respect to the predicted contribution to the photonic band gap. The remaining part of the centrifuged LNPs was classified into different layers in the pellet according to their particle size. By DLS we con-firmed that each layer of the centrifuged LNPs has a narrow PDI (<0.05), which is important for their close-packing.

We observed a red-to-violet rainbow coloration across the cross-section of the crystal collected from the pellet, with violet color on the surface. This coloration remained stable after drying the pellet under ambient temperature. These results indicate that LNPs interact with light because the spacings of the ordered regions of LNPs fall within the wavelength region of visible light.

### Rectangular patterns of LNPs formed by evaporation-induced self-assembly

Crystallization from colloidal dispersion to crystal phases has been observed when increasing the particle concentration[13] for example by gravitational sedimentation[7] or vertical deposition[6]. However, mate-rials with a photonic band gap do not always form spontaneously upon reaching a threshold concentration in the colloidal dispersion[37]. For example, when we let a dispersion of LNPs evaporate in a Petri dish at room temperature, we observed the formation of rectangular platelets (Fig. 2b, c). Although such ordered platelet stripe patterns were formed by evaporation-induced self-assembly no structural color was observed (Fig. 2b). Due to the absence of a photonic band gap the platelets remained brownish in accordance with the original lignin color. A higher magnification scanning electron microscopy (SEM) image of the surface of a single platelet revealed that it consists of a rather polydisperse distribution of particles (Fig. 2e). In addition to showing some unfilled spots within the surface the average size of LNPs was 77 nm, with standard deviation of 20 nm, and PDI of 0.07 (Supplementary Fig. 2). In addition to the PDI exceeding 0.05 the particle size was below that predicted to form structural colors (Table 1). Nevertheless, the ordered rectangular structure provided

**Table 1 | Particle size (D) dependency of structural color cal-culated from a modified Bragg's equation**

|            | Violet  | Blue    | Green   | Yellow  | Orange  | Red     |
|------------|---------|---------|---------|---------|---------|---------|
| $\lambda$ (nm) | 380–450 | 450–495 | 495–570 | 570–590 | 590–620 | 620–750 |
| $D$ (nm)   | 158–187 | 187–205 | 205–236 | 236–245 | 245–257 | 257–311 |

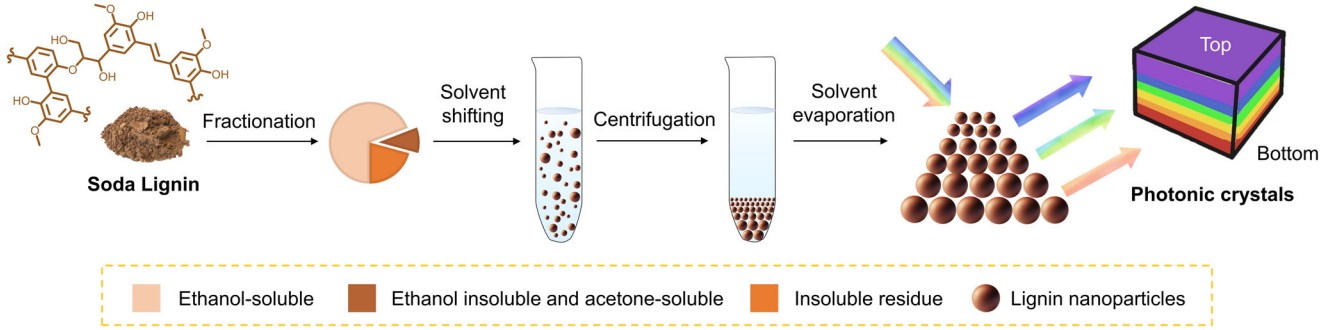

**Fig. 1 | Scheme of preparation of photonic crystals from lignin.** Solvent fractionation of soda lignin and preparation LNPs by a solvent shifting method. After centrifugation, the layers of the rainbow-colored crystal form according to the particle diameter. Note, not drawn to scale.

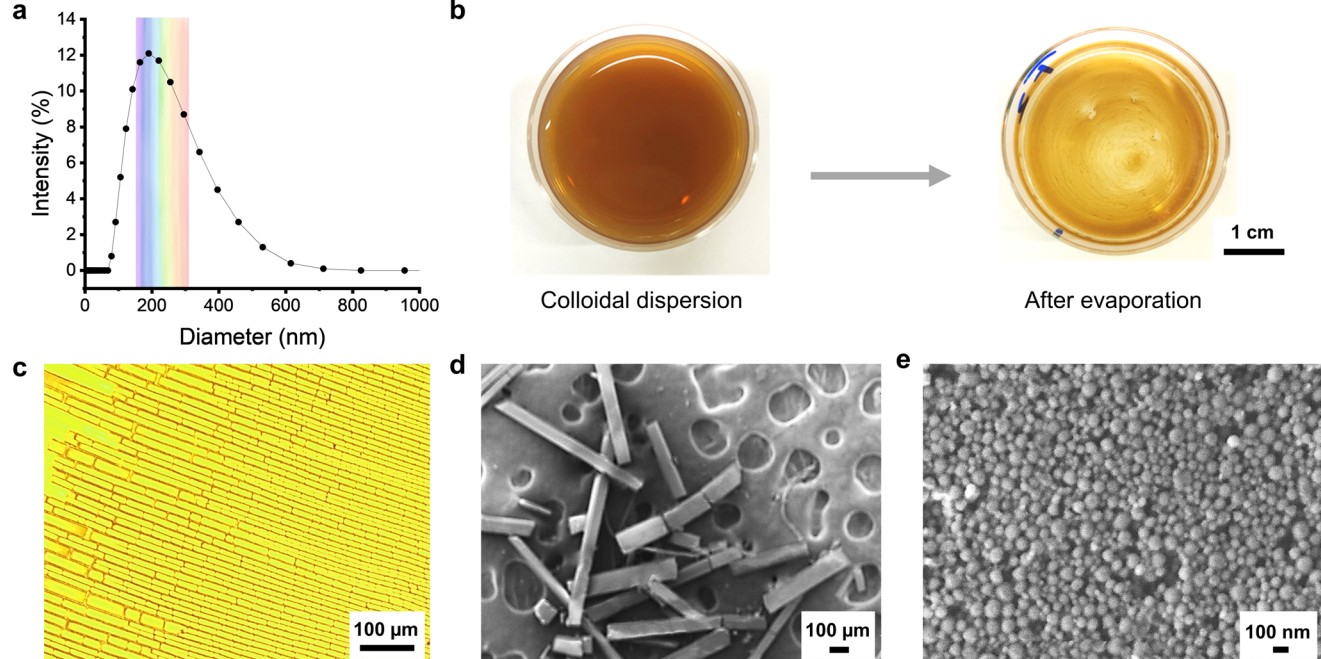

**Fig. 2 | Formation of stripe pattern of LNPs by evaporation-induced self-assembly. a** Size distribution of LNPs based on DLS. **b** Digital photos of Petri dish before and after evaporation. **c** Optical microscope image of stripe patterns. **d** and **e** SEM images of platelets and a closer view of a single platelet surface.

hints towards the ability of LNPs to assemble into close-packed domains and suggest that the process may require size-dependent sedimentation and particle classification.

## Structural color of LNPs formed by centrifugation-assisted assembly

Since evaporation-induced self-assembly only works in forming platelet patterns, we hypothesized that the centrifugation of LNPs could assist the formation of ordered structures. There are two important effects that centrifugation may have on the formation of ordered close-packed structures. Firstly, centrifugation provides a rapid way to increase the volume fraction of LNPs. This is important from a theoretical foundation point of view that considers that the threshold volume fraction that needs to be overcome to achieve phase transition is 0.494[13]. Secondly, centrifugation classifies LNPs based on their hydrodynamic diameter, which is important for achieving monodisperse subdistributions. After centrifugation, we observed a brownish supernatant and a colorful pellet in a wet state (Supplementary Fig. 3a). The brownish color of the supernatant comes from small LNPs which did not sediment. Violet is the main color on the surface of the pellet, while also some blue and green coloration can be

observed on the edges. We removed the supernatant and dried the pellet at room temperature. We observed gradient colors after complete solvent evaporation (Supplementary Fig. 3a). The photonic structure was revealed by SEM. As shown in Fig. 3a, some regions of LNPs are semi-hexagonally close-packed. However, the stability of the structure might be disrupted after sectioning as evidenced by a scanning transmission electron microscope (STEM) image of a 200 nm-thick section from the LNPs photonic crystal, which showed that some particles had become detached, possibly due to weak interparticle interactions of LNPs (Supplementary Fig. 4). Future studies can investigate the stability of lignin photonic crystal structures in processing applications, particularly their resistance to mechanical and solvent stresses.

The violet color was on the top surface of the dried pellet and the red color is on the bottom surface. In between, we observed a rainbow-type color gradient. The order of the colored layers matches that of the predicted particle size ranges (Table 1). The red layer was on the bottom because it originates from the largest particles in the population, which sediment first during the centrifugation. In turn, small particles are on the top because they have a slower rate of sedimentation and thus contribute to the violet

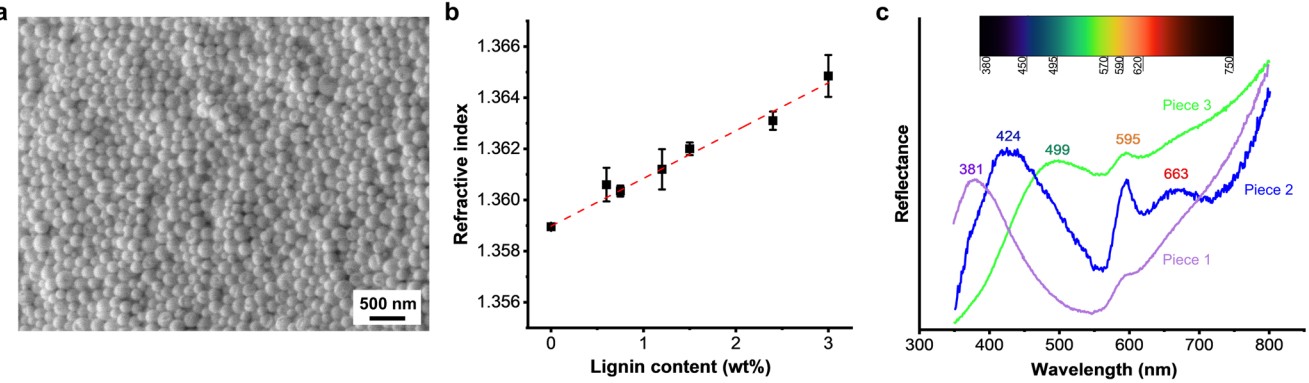

**Fig. 3 | Structural and optical properties of lignin and LNPs photonic crystal.** **a** SEM image of the surface of photonic crystals. **b** Refractive index of lignin solution in acetone. Dashed line represents a liner fit (intercept = 1.3590; slope = 0.00187; Pearson's $r$ = 0.9949). Error bars represent standard deviation based on the entire population ($n$ = 3). **c** Reflectance spectra of the photonic crystals from three distinct pieces with different colors of violet, blue, and green.

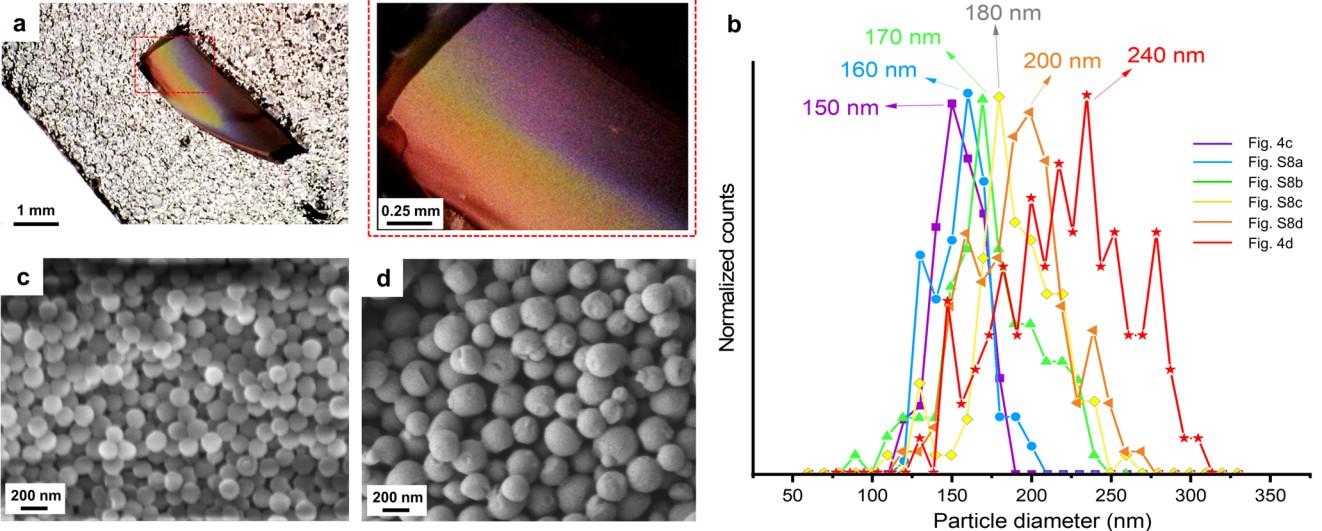

**Fig. 4 | Characterization of the structure of LNPs photonic crystal at different color layers.** **a** Digital microscope image of a cross-section of a photonic crystal. **b** Size distributions of LNPs in different color layers of the photonic crystal. More than 100 particles were measured in SEM images from **c**, **d** and Supplementary Fig. 8. The peaks are at 150, 160, 170, 180, 200, and 240 nm, respectively. SEM images of the fracture cross-section of the photonic crystal in violet (**c**) and red (**d**) layers.

photonic band gap. We measured the refractive index of lignin by dissolving different amounts of lignin in acetone (Fig. 3b). The refractive index of lignin was calculated to be 1.54 by extrapolating from a fitting line ($y = 0.0018x + 1.3591$, $R^2 = 0.98$) and found to be similar as previously reported[38].

In agreement with our visual observations, UV–vis reflection spectra revealed peaks in various wavelength regions of visible light. The peaks at 381, 424, 499, 595, and 663 nm are responsible for violet, blue, green, orange, and red, respectively (Fig. 3c). No sharp peaks were observed in the yellow light range, which might be because yellow light has the narrowest wavelength range (570–590 nm) and correspondingly only a low amount of LNPs fell to the corresponding particle size range (236–245 nm, Table 1). Another reason might be the strong adjacent peaks from orange and green colors. The reflectance of the photonic crystal was plotted in a chromaticity diagram as shown in Supplementary Fig. 5. The structural color originates from a coherent scattering of light within the ordered photonic structures[33]. In addition, lignin and LNPs did not show the ability of absorbing specific wavelengths of visible light (Supplementary Fig. 6), which indicated that they are different from light-absorbing pigments.

## Size distribution of different color layers

The size-dependent structural color of the photonic crystals was predicted by theoretically calculated particle diameters of LNPs (Table 1). Here we further confirm the trend of size dependence by SEM imaging and compare the experimental results with the theoretical values. To image samples with SEM, we sputter-coated the surfaces of the samples with gold. No change in color was observed after coating compared to the digital microscope image before coating (Supplementary Fig. 7). As shown in Fig. 4a, the cross-section of LNPs photonic crystal exhibits a gradient color from violet to red. The thickness of the LNPs photonic crystal was 1.1 mm (accuracy: ±0.1 mm). By imaging different color layers across the photonic crystal, we collected corresponding SEM images, as shown in Fig. 4c in violet, Fig. 4d in red, and in Supplementary Fig. 8, blue, green, yellow, and orange. The size distributions of LNPs in each color layer were more uniform and monodispersed than in Fig. 2e which is the image of LNPs after gravitational sedimentation. Image analysis of the SEM micrographs gave the size distributions of LNPs in different layers (Fig. 4b). The peaks of particle size distributions were located at 150, 160, 170, 180, 200, and 240 nm, from violet to red, respectively. The size-dependence of LNPs photonic crystal agrees with the

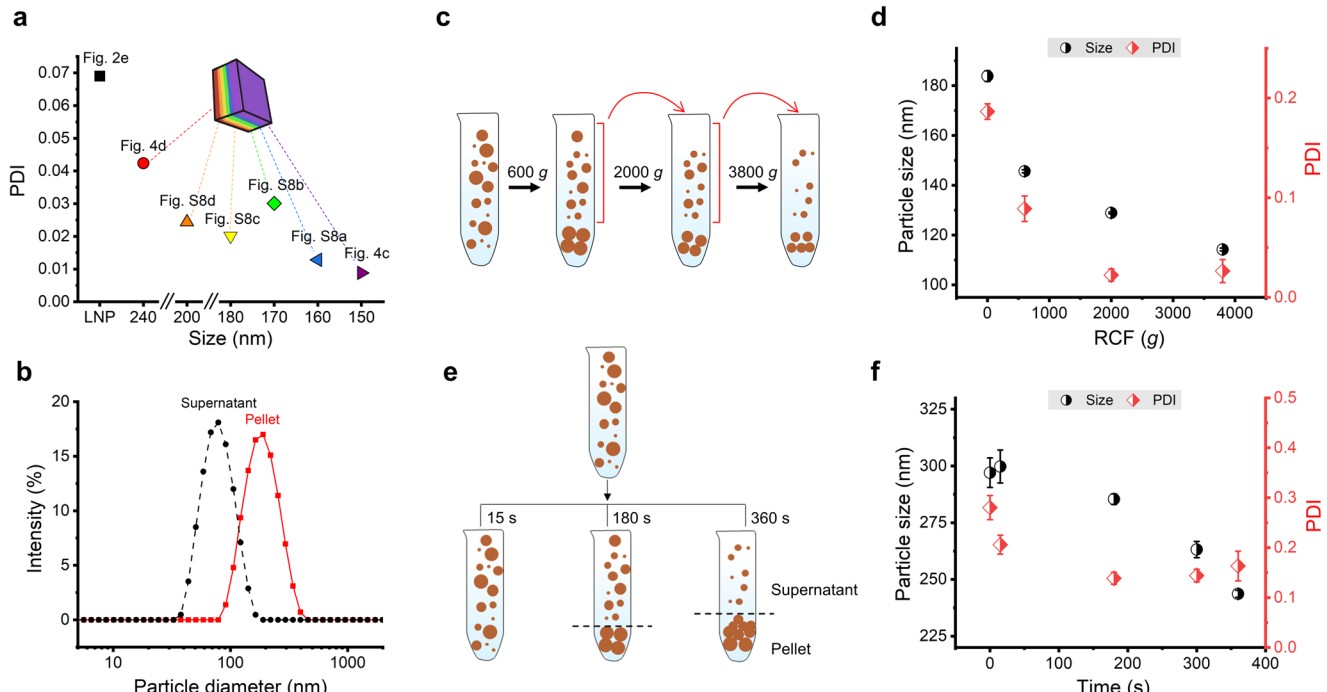

**Fig. 5 | Mechanism study of size separation. a** Calculated polydispersity index (PDI) of LNPs in different color layers formed in the pellet after centrifugation. The legend indicates the source images used in the image analysis. Calculation is presented in Supplementary Table 1. **b** DLS size distribution of LNPs in pellet and supernatant after centrifugation at 3800×g for 2 h. **c** Scheme of differential centrifugation. **d** Particle size (diameter) and PDI of supernatant after differential centrifugation with different RCF, first run with RCF of 600×g, second 2000×g, and third 3800×g, from DLS results. Error bars represent standard deviation based on the entire population (n = 3). **e** Scheme of centrifugation for different times. **f** Particle size (diameter) and PDI of supernatant after different centrifugation times, from DLS results. Error bars represent standard deviation based on the entire population (n = 3).

photonic theory, that is, a reduction in size induces a blue-shift when the refractive index and packing structure remain constant. The particle diameter of LNPs for different colors acquired from SEM images is smaller than predicted in Table 1. This difference might be because, despite their structural color, LNPs were not perfectly close-packed in cubic domain therefore the real inter-planar spacing $d_{111}$ might be slightly higher than predicted from Eq. (2). After these insights we transferred our attention to elucidate the mechanism by which LNPs form ordered photonic structures.

**Elucidating the effect of centrifugal forces on hierarchical particle assembly**

Narrow size distribution is considered important in forming colloidal crystals[39]. In line with this foundation, we found that the PDI of LNPs in all layers of the photonic crystals was lower than 0.05. As shown in Fig. 5a, the PDIs (see Supplementary Table 1) of the layers corresponding to red to violet colors were between 0.008 and 0.043, which is markedly lower than the PDI of initial LNPs (0.07, Supplementary Fig. 2). The monodispersity increased by centrifugation is important to the formation of the photonic crystal. We see in Fig. 5b that centrifugation can separate LNPs in supernatant and pellet according to size, 70 and 190 nm, PDI 0.08 and 0.18, respectively. Theoretically, particles with a diameter of 70 nm are too small to generate a photonic band gap (Table 1). Therefore, removing the supernatant enriched with the smaller particles helps to avoid the formation of defects in the photonic crystals.

The size separation in the process of centrifugation can be studied by varying centrifugation force and time. Relative centrifugal force (RCF) can be converted from revolutions per minute (rpm), using the following formula: RCF = (rpm)2 × 1.118 × 10 − 5 × r (r: centrifugation radius in cm). Based on Stoke's law, the velocity ($v$) of sedimentation can be calculated from: $v = \frac{D^2 (\rho_{Particle} - \rho_{Liquid}) \times g}{18\mu}$, therefore the velocity

depends on five parameters. When the density of particles ($\rho_{Particle}$), the density of the liquid ($\rho_{Liquid}$), gravitational force ($g$), and solvent viscosity ($\mu$) are constant, particle size ($D$) is crucial to the sedimentation speed, which can be proved by differential centrifugation experiments. As shown in Fig. 5d, the initial hydrodynamic size of LNP was 183 nm, PDI 0.19. After centrifugation with RCF of 600×g, the particle diameter in the supernatant decreased to 146 nm, which means that large particles in the initial dispersion participated in the formation of the pellet. Then the supernatant was centrifuged in a second run with RCF of 2000×g, and the second supernatant showed a diameter of 129 nm. After the third run of centrifugation with RCF 3800×g, the particle diameter in the supernatant slightly decreases to 114 nm. We can conclude that bigger particles have a faster rate of precipitation; to allow smaller particles to participate in the sedimented phase extended duration of centrifugation or higher RCF is needed. The influence of centrifugation time is shown in Fig. 5f by carrying out several parallel experiments. Here we chose LNPs with a larger hydrodynamic size of ~300 nm and a broader PDI of 0.28 (Supplementary Fig. 1) to demonstrate the time influence of centrifugation. As shown in Fig. 5f, the average particle diameter in the supernatant decreases when a longer centrifugation time is applied. After a short centrifugation time of 15 s, no significant change was observed in the size distribution. After 180 s centrifugation, the particle size in the supernatant decreased to 285 nm, which means that large particles in the initial dispersion formed the pellet. The particle size decreased to 263 nm after 300 s and further to 244 nm after 360 s of centrifugation, which means that both large and medium-sized particles were deposited into the pellet. As shown in the illustration of Fig. 5e, after centrifugation for 360 s, the size gradient from large to small occurs from bottom to top in the pellet, which corresponds to the SEM results in Fig. 4. Therefore, narrow size distribution is achieved in different layers after centrifugation.

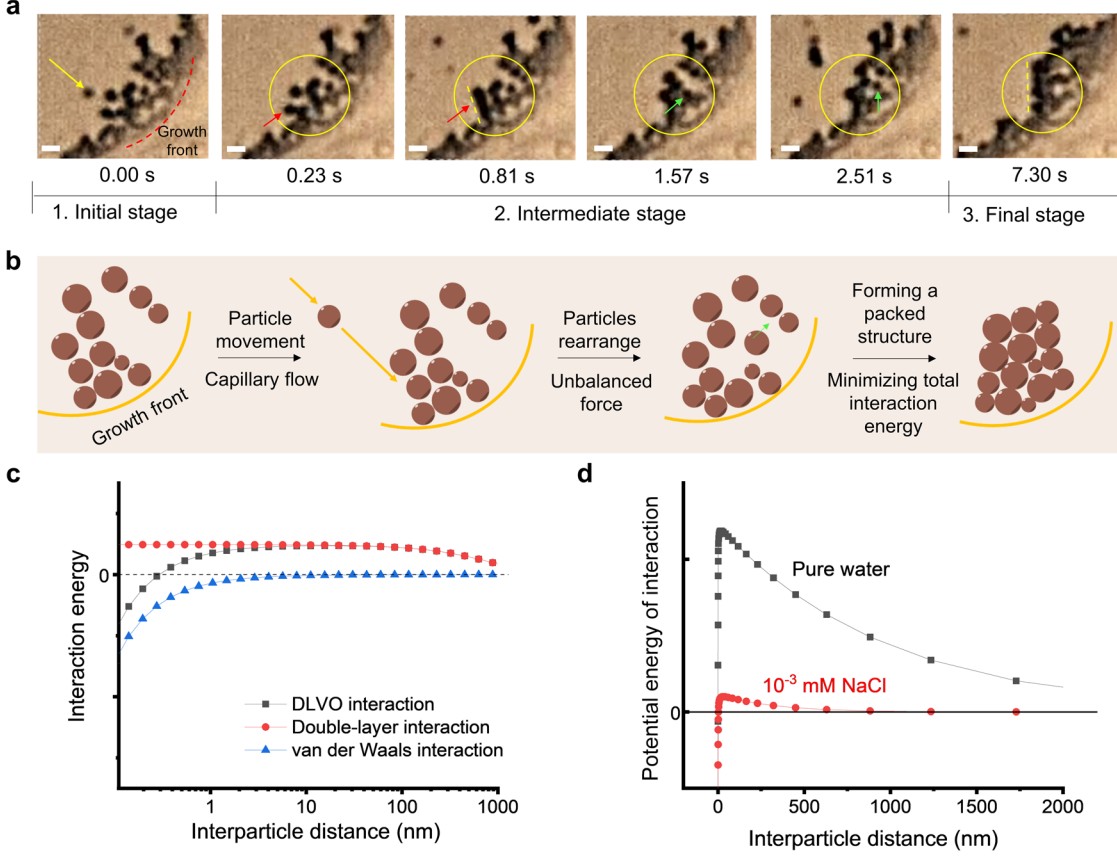

**Fig. 6 | Dynamic analysis of particle assembly and interactions. a** Microscopy images from movie recordings captured during evaporation-induced self-assembly. The growth front is shown by a red dashed line. Scale bar 2 μm. Red and green arrows indicate two particles and how they rearrange. **b** Schematic illustration of the formation of a packed structure via rearrangement of particles. **c** Calculations of interactions (repulsive double-layer interaction, attractive van der Waals interaction, and DLVO interaction) in LNPs colloidal system by using the DLVO model. **d** Calculations of DLVO interaction of LNP particles in pure water and in the salt solution.

## Understanding the dynamic self-assembly process

We assume that the rearrangement of particle assemblies can occur as long as there is a driving force. For easier observation, we prepared LNPs with a hydrodynamic size of 582 nm (Supplementary Fig. 1) and studied their dynamic assembly during evaporation by in situ microscopic observation (Supplementary Movie 1). As shown in Fig. 6a, chronicled screenshots from the movie recording reveal how particles form a packed structure. The growth front[40], that is, from wet to dry regime, is shown by a black dot curve at the right bottom corner of the first image (0 s). Initially (0 s), a single particle which is pointed by a yellow arrow was brought to the edge of the evaporation front by capillary forces. This movement of the particle induces an unbalanced force on the particles which exist at the growth front. As shown in the images at 0.23 and 0.81 s, another particle which is pointed by a red arrow started to move from its original position, and at the end, the particles formed a more packed structure, as shown by a yellow dot line in the image of 0.81 s. A similar phenomenon was also observed from 1.57 to 7.30 s. As pointed by a green arrow, a particle dynamically rearranged from its original position to a new position. At the end, a packed structure formed, as shown with a yellow dot line in the image recorded at 7.30 s. These observations explain the self-assembly of LNPs to form rectangular platelets in Fig. 2c and d.

For self-assembly to occur, a balanced combination of driving forces and colloidal forces is necessary[8]. Our in situ observation provides a testimony that particles experience three stages during rearrangement and that their assembly is driven by the minimization of total interaction energy. At the initial stage, particles were brought to the edge of the evaporation front via capillary-induced water flow[40-42].

The movement of new incoming particles brings an unbalance force to the existing particles which are already at the growth front, therefore these particles rearrange to find another position where they are force-balanced. We call this step the intermediate stage. The DLVO (Derjaguin, Landau, Verwey, and Overbeek) theory[35] which describes the interaction forces between charged particles can explain the unbalanced force (Fig. 6c and d). As shown in Fig. 6d, there is a peak in the DLVO potential when two particles are brought near, for example, from 1 μm to 10 nm. This strong repulsive peak is due to the charged surface of LNPs and the long Debye screening length in a low ion concentration in an aqueous solution (Fig. 6d). At the final stage, particles rearrange until they form sufficiently large packed structures. This forming process is driven by minimizing total interaction energy[43].

## Discussion

We have elucidated differences between evaporation-induced and centrifugation-assisted sedimentation of lignin nanoparticles into packed structures. Rectangular platelets resulted from evaporation under ambient conditions while photonic crystals appeared upon centrifugation. The relation between the particle size and observed structural color was derived from a modified Bragg's equation. Microscopic investigation revealed the ordered packing of lignin nanoparticles in the photonic layers that displayed a color gradient from violet to red as observed by the naked eye and confirmed by reflection spectra in the visible wavelength range. Centrifugation has several key contributions to structure formation. It removes small particles which disturb the photonic structures, increases the solid

content of the pellet, classifies the particles, and increases monodispersity in each sedimented layer. Within the photonic layers, the particles were highly monodispersed (PDI < 0.05) and classified according to their physical size ranging from 150 to 240 nm. The in situ microscopy images provided additional insight into the dynamic rearrangement, assembly, and interparticle interactions of lignin nanoparticles. We conclude that the formation of photonic, close-packed ordered structures can be achieved by centrifugation-assisted assembly of lignin nanoparticles, which opens avenues towards advanced functional materials from these abundant and low-cost building blocks.

## Methods

### Materials and chemicals

All prepared lignin materials (LNPs and LNPs photonic crystal) in this work were prepared from soda lignin (PROTOBIND 2400, GreenValue LLC, purity: 90%, previously characterized by $^{31}$P NMR spectroscopy[36]). Ethanol (purity: 95%) from Kiilto, Sweden, and acetone (purity: 99.5%) from Honeywell, Germany were used as solvents in the fractionation procedure. Deionized (DI) water was used in all experiments. All chemicals were used as bought unless noted.

### Fractionation of soda lignin

Lignin was first purified with ethanol and then extracted with acetone[20]. Specifically, 50 g soda lignin was added to 400 mL ethanol under 600 rpm string for 30 min. The solid part of the lignin suspension was collected by centrifugation (RCF: 3800×$g$, 10 min) and the solution part was removed. 300 g acetone was added to the solid part of lignin and then strung under 600 rpm for 1 h. The suspension was filtrated and the fractionated solution was used for preparing LNPs for the next step.

### Preparation of lignin nanoparticles (LNPs)

LNPs were prepared by solvent shifting method[36]. The fractionated lignin solution (solid content: ~5 g) was added to 100 g DI water and stirred under 600 rpm for 1 h. Then 1200 mL water was added to the solution (acetone/water, 3/1, w/w) with a water dilution rate of 20 v/v/min, LNPs size was around 190 nm, and the final LNP dispersion was 0.18 wt%. Another two different size LNPs (316 and 582 nm) were prepared by varying water dilution rates to 3 and 0.25 v/v/min, respectively.

### Evaporation-driven assembly of LNPs into platelets

The stripe pattern was prepared by evaporation-induced assembly of LNPs. 3 mL LNPs dispersion was added to a Petri dish (diameter: 3.5 cm). The pattern was formed after solvent evaporation.

### Preparation of LNPs photonic crystal

LNPs photonic crystal was prepared by centrifugation (Sorvall LYNX 6000 Superspeed Centrifuge). Specifically, 200 mL LNPs dispersion (size: ~190 nm) was added to 500 mL centrifuge bottle and centrifuged at the force of 3800×$g$ for 2 h. The supernatant was carefully removed and the pellet was dried at room temperature.

### Dynamic light scattering

A Zetasizer Nano ZS (Malvern, UK) was used to measure the size distribution of LNPs by dynamic light scattering (DLS). 20 μL of LNPs dispersion was added to 2 mL DI water. The diluted LNPs dispersion samples were measured 3 times with 12 runs per run. Median values were reported.

### Refractometry

A refractometer (ATAGO, Abbe Refractometer NAR-1T SOLID) was used to measure the refractive index of lignin in acetone solution at 21 °C.

### UV–Vis spectrophotometry

A UV–Vis spectrophotometer (Agilent Cary 5000 UV–Vis–NIR) was used to measure the optical properties of LNPs photonic crystal in the visible range. The reflectance spectrums of LNPs photonic crystal were recorded with the praying mantis diffuse reflectance accessory in the range of 350–800 nm with a specialized fluorine-based polymer as reference. And the beam spot size was 1.5 mm. A UV–Vis spectrophotometer (Thermoscientific Genesys 150) was used to measure the absorption spectra of lignin and LNPs. Lignin solution and LNPs dispersion were coated on a cover glass (borosilicate glass, 24 × 32 mm), respectively, and their absorption spectra were recorded. The spectrum of cover glass was recorded as a blank.

### Microscopy

A JSM-7401F field emission scanning electron microscope was used to image LNPs and LNPs photonic crystal samples. An accelerating voltage of 1–2 kV and a working distance of 2–15 mm were used during measurement. Some samples were coated for 60–120 s with gold using a JFC-1200 fine coater before the SEM study. The gold particles added by sputtering are about 5–15 nm in size and they were added to increase the contrast in the images. Annular dark field (ADF) scanning transmission electron microscopy (STEM) images were obtained using a ThermoFisher Themis $Z$ double aberration-corrected TEM operated at 300 kV with a convergence angle of 21 mrad and a dwell time of 3 μs. The material was sectioned to 200 nm thickness using ultramicrotomy using a Leica Ultracut UCT with a 45° diamond knife (Diatome) after the crystalline lignin nanoparticles first had been embedded in Agar low viscosity resin to facilitate sectioning.

A Nikon FN-S2N (Japan) microscope was used to image the rectangular platelets of LNPs and record a movie of the assembly and rearrangement of LNPs during evaporation. A Dino-Lite Edge 3.0 digital microscope was used to image the photonic crystals of LNPs.

### Calculation of particle size–wavelength relationship

The particle size which responds to different visible wavelength scales (at normal incidence) can be predicted from the diffraction of electromagnetic waves following a modified Bragg's law[7,34,35].

$$\lambda = 2n_{\text{eff}} \, d_{111} \tag{1}$$

where $n_{\text{eff}}$ is the effective refractive index of lignin, $d_{111}$ is the interplanar spacing within the (111) plane, the latter dependent on $D$ which is the particle diameter (unit: nm), for an array of cubic close-packed spheres, and under the assumption that particles are rigid and touching, the relationship between the sphere radius and the spacing of the (111) planes is

$$d_{111} = \sqrt{\frac{2}{3}} D \tag{2}$$

the $n_{\text{eff}}$ can be calculated by using the following equation:

$$n_{\text{eff}} = \varphi n_{\text{lig}} + (1 - \varphi) n_{\text{air}} \tag{3}$$

$n_{\text{lig}}$ and $n_{\text{air}}$ are refractive indices of lignin and air, 1.6[38] and 1, respectively, and $\varphi$ is the volume fraction of particles in a close packing, which is 0.7405.

### Interparticle force calculations based on the DLVO theory

Repulsive double-layer interaction $V(D)_{\text{DL}}$ can be calculated by Eq. (4)[35]:

$$V(D)_{\text{DL}} = \frac{64\pi RkTn(\infty)}{\kappa} \tanh 2\left(\frac{z\phi_0}{4kT}\right) \exp(-\kappa D) \tag{4}$$

where $R$ is particle size ($\approx$ 500 nm), $k$ is Boltzmann constant (=$1.38 \times 10^{-23}$ J/K), $T$ is absolute temperature (= 300 K), $n(\infty)$ is number density of ions in bulk solution (for pure water, = $6.022 \times 10^{19}$ ions/m$^3$), $z$ is ion valency (for pure water, = 1), $\phi_0$ is surface potential ($\approx -37$ mV [36]), $D$ is the distance between particles. $\kappa^{-1}$ is Debye-length, which can be calculated by

$$\kappa - 1 = \sqrt{\frac{kT\varepsilon_0\varepsilon_r}{2e^2z^2n(\infty)}} \qquad (5)$$

where $\varepsilon_0$ is permittivity of vacuum (= $8.854 \times 10^{-12}$ F/m), $\varepsilon_r$ is relative dielectric constant (=78.5 F/m), $e$ is elementary charge (=$1.6 \times 10^{-19}$).

Attractive van der Waals interaction $V(D)_{vdW}$ can be calculated by

$$V(D)_{vdW} = -\frac{AR}{12D} \qquad (6)$$

where $A$ is Hamaker constant (=$1.7 \times 10^{-20}$ J[44]). The force is always attractive between two identical materials across a medium.

Therefore, DLVO interaction $V(D)_{DLVO}$ can be calculated by

$$V(D)_{DLVO} = V(R) + V(A) \qquad (7)$$

## Data availability

All data generated in this study have been deposited in the Zenodo repository database under accession code 10.5281/zenodo.7860969. All data generated in this study are provided in the Source Data file.

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

## Acknowledgements

The authors acknowledge funding from the Knut and Alice Wallenberg Foundation (KAW) through the Wallenberg Wood Science Center (grant number KAW 2018.0452). T.W. acknowledges support from the Swedish Research Council (VR, 2019-05465). Funded by the European Union (ERC, CIRCULIG, Project 101075487). Views and opinions expressed are however those of the author(s) only and do not necessarily reflect those of the European Union or the European Research Council Executive Agency. Neither the European Union nor the granting authority can be held responsible for them.

## Author contributions

J.L. and M.H.S. designed the study. J.L., M.N., K.J., and T.W. performed the experiments. J.L., M.N., K.J., T.W., and M.H.S. interpreted the data. J.L. and M.H.S. wrote the first draft. All authors contributed to revising and writing the final version of the manuscript.

## Funding

## Competing interests

The authors declare no competing interests.
