## [Peer Review File · Nature Communications]

Photonic crystals with rainbow colors by centrifugation-assisted assembly of colloidal lignin nanoparticlesREVIEWER COMMENTS

Reviewer #1 (Remarks to the Author):

The author described a PhC with rainbow colors fabricated from lignin nano particles. What is the size distribution of the lignin particles? Lignin particles have different size, so the Phc show rainbow colors? The PhC was assembled by centrifugation, a SEM for the cross-section of PhC is desired.

Reviewer #2 (Remarks to the Author):

Based on lignin nanoparticles, the photonic crystals with structural colors was constructed by a centrifugation-assisted method. The preparation mechanism and formation process of photonic crystals were deeply analyzed. The article has some innovation, but the following problems need to be solved and explained:

1. The abstract is poorly written;
2. The innovation point is not clear enough, there are many renewable substances, why only choose lignin?
3. What are the advantages of centrifugal-assisted fabrication of photonic crystals?
4. The schematic drawing is not graphic enough. Why not use a spherical shape for the LNP in Figure 1?
5. What is the thickness of the photonic crystal and how to control it?
6. What is the stability of the structure of lignin photonic crystals prepared by centrifugal-assisted method?
7. What are the potential applications of lignin photonic crystals?
8. What is the meaning of the particle sizes of lignin nanoparticles in Table 1? Hydrated particle sizes or dried particle sizes?
9. Compared with the References named "Monodispersed Lignin Colloidal Spheres with Tailorable Sizes for Bio-Photonic Materials" and "Photonic Lignin with Tunable and Stimuli-Responsive Structural Color", what are the advantages of this paper?
10. The structure colors of the prepared photonic crystals are not bright enough, and some necessary color characterization means are lacking;
11. The axes in Figure 5 are calibrated either inward or outward;
12. Some of the pictures in this article are not clear enough.

Reviewer #3 (Remarks to the Author):

[Note from the editor: Please also see attached PDF]

Manuscript: Photonic crystals with rainbow colors by centrifugation-assisted assembly of colloidal lignin nanoparticles

Summary of content:

Liu et al. demonstrate a centrifugation-assisted assembly of lignin nanoparticles. Lignin is a bio-polymer which the authors synthesize into different nanoparticle sizes. After self-assembly, structural color is emitted from the structure covering the whole optical range. A study of the self-assembly dynamics is also provided.

Comments to the Manuscript:

Generally, the manuscript is well written apart from a few typos (see below) and due to its social relevance, I recommend the manuscript to be accepted after major revisions.

In the introduction, I am missing a comparison of advantages and disadvantages compared to cellulose based optics.

e.g. <https://onlinelibrary.wiley.com/doi/10.1002/adma.202109170>

I would recommend to remove the bracket from the following sentence in the introduction as it does not help the understanding.

Visible coloration results when the lattice constant (the lengths a , b , and c of the 2 three cell edges meeting at a vertex, and the angles α , β , and γ between those edges) of the crystal is comparable or smaller than the wavelength of light and the bandgap corresponds to a wavelength in the visible range (400–800 nm).

On page 4 of the result section, the abbreviation of the unit $v/v/\text{min}$ should be once written out.

In the context of discussing the optics, a measured refractive index and absorption of a lignin thin film would be very valuable.

In a few subfigures, the legends are missing. These are Fig. 3b, Fig. 5a,b,d,f. In general, the authors should pay attention to the color blind people. There are two websites where color compatibility can easily be checked.

Color Scheme Designer: <http://89.161.193.92/colordesigner/>

Color Brewer: <https://colorbrewer2.org/#type=sequential&scheme=BuGn&n=3>

Figure 3 and 4 are to a certain extent doubled. I would recommend to discuss the optical response in a separate figure with the measurements of refractive index. The beam spot size of the UV-Vis reflection spectra should be given. The image of the rainbow film is shown three times but is not providing extra information. I would restrict the presentation to Fig.5a. Fig. 3c does not add any information. Fig. 4 b should be larger to allow for better reading. The SEM images in 4f and g should either be repeated in higher quality or moved to the SI.

The phrase The yellow peak is unclear is not really scientific and should be better phrased.

The calculation of the PDI of LNPs should be presented in the SI. As the shapes of the curves are really rough, it would be valuable to know how the data was treated. Were there any fits applied? Was the PDI only extracted from the one sharp sub-peak at highest intensity etc? How many NPs have been analyzed.

Comment to Figure 5b: The r and n in supernatant are very close together and appear nearly as a m .

In the microscopy section in the characterization, it is mentioned that a STEM was used but there is no single STEM image neither in the manuscript nor in the SI.

Summary:

In summary, the content of the manuscript is relevant for a substantial part of Nature Communications target readership. The relevance of sustainable photonic components is very high. The presentation of scientific findings is consistent. Before publication, the discussion of the optical properties should be extended and the Figures 3 and 4 of the characterization should be better organized.

Typos and small remarks:

Abstract:

-In situ optical microscope images provided additional evidence on the importance of dynamic rearrangement of LNPs during their assembly into semi-closed packed structures.

Introduction:

-The brown-black color palette of industrial lignin limits the opportunity window for optical applications.

Results:

-To answer this question, we can look into the origin of this phenomenon.

-The size-dependent structural color of the photonic crystals was predicted by theoretically calculated particle diameters of LNPs (Table 1).

-To image samples with SEM, we sputter coated the surfaces of the samples with gold, which did not change their color as observed color appearance (SI Fig. S3)

Comment: Please rephrase the yellow indicated part, to make the meaning clearer.

-RCF explain abbreviation at the first occasion at which it occurs.

-DLVO abbreviation should be written out at the first occasion at which it occurs.

Methods:

-Lignin was first purified with ethanol, and then was extracted with acetone, according to a method reported in the literature.

-LNPs were prepared by similar solvent shifting method from literature.

-The fractionated lignin solution (solid content: ~ 5 g) was added to 100 g DI water and stirred under 600 rpm for 1 h.

Responses to Reviewer's comments:

We would like to thank the expert reviewers for their critical comments on our manuscript, which allowed us to achieve the current improved version. We have taken into account all the insightful comments of the reviewers and revised the manuscript accordingly. Our point-by-point responses are included in blue font after each reviewer comments that are available in black. The changes made to the manuscript are shown in red color. A clear revised version of the manuscript and a red-marked version with all changes clearly indicated are included in this submission.

Reviewer#1 (Remarks to the Author):

The author described a PhC with rainbow colors fabricated from lignin nano particles. What is the size distribution of the lignin particles? Lignin particles have different size, so the Phc show rainbow colors? The PhC was assembled by centrifugation, a SEM for the cross-section of PhC is desired.

Our response: Thank you very much for reviewing our manuscript, and for arising these questions and suggestion. The size distribution of lignin particles in different layers are from 150 to 240 nm, please see figure 4b below:

Figure 4. (b) Size distributions of LNPs in different color layers of the photonic crystal. More than 100 particles were measured in SEM images from (c), (d) and Supplementary Fig. 8.

Yes, we found that the photonic crystals show rainbow colors because of the different sizes.

We appreciate the review's suggestion. We tried to image an ion polished cross-section but the sample was melted. Therefore we have added the SEM for the fracture cross-section of PhC in supporting information, as follows:

Figure S8. SEM images of the fracture cross-section of LNPs in different color layers: (a) blue, (b) green, (c) yellow, and (d) orange.

Reviewer#2 (Remarks to the Author):

Based on lignin nanoparticles, the photonic crystals with structural colors was constructed by a centrifugation-assisted method. The preparation mechanism and formation process of photonic crystals were deeply analyzed. The article has some innovation, but the following problems need to be solved and explained:

Our response: Thank you very much for reviewing our manuscript, highlighting its strengths and shortcomings.

1. The abstract is poorly written;

Our response: We have rephrased the abstract. Please see the revised manuscript.

2. The innovation point is not clear enough, there are many renewable substances, why only choose lignin?

Our response: We agree that it is needed to clarify the motivation of our choice of lignin. We have highlighted our innovations and motivations in the revised manuscript as follows: "Cellulose, hemicellulose and lignin are the main components of plant biomass. The structural color of cellulose-based optics comes from helicoidal assembly structure of cellulose nanocrystals, and there has been significant progress in understanding the key processes involved¹⁵⁻¹⁹. The development of lignin-based structural colors is still in its early stages^{20,21} and further understanding of their formation and properties is highly desired. This advancement is important to pave the way for value-added lignin photonic materials, given that the current practice is to combust lignin in the chemical recovery process of kraft pulp production²²." Please see the revised introduction for other minor changes.

3. What are the advantages of centrifugal-assisted fabrication of photonic crystals?

Our response: We have highlighted the advantage of centrifugal-assisted fabrication of photonic crystals in the revised manuscript as follows: "The advantages of centrifugation are to classify LNPs according to their particle size and produce monodispersed particle layers that displayed gradient colors from red to violet."

4. The schematic drawing is not graphic enough. Why not use a spherical shape for the LNP in Figure 1?

Our response: We appreciate the reviewer's suggestion. We have changed Figure 1 accordingly, as follows:

5. What is the thickness of the photonic crystal and how to control it?

Our response: We have added the thickness data in the revised manuscript when discussing figure 4, as follows: "The thickness of the LNPs photonic crystal was 1.08 mm." The thickness can potentially be controlled by changing the concentration of the lignin dispersion, but we do not have unambiguous data to prove it at this point."

6. What is the stability of the structure of lignin photonic crystals prepared by centrifugal-assisted method?

Our response: Thank you very much for raising this question. The structure is stable after evaporation as shown in SI Fig. S3b (below with highlighted examples of these regions) and Fig. S3c. The stability might be decreased after sectioning as shown in SI Fig. S4. We have rephrased and discussed the stability issue more in depth in the revised manuscript, as follows: "We observed gradient colors after complete solvent evaporation (SI Fig. S3a). The photonic structure was revealed by SEM. As shown in Fig. 3a, some regions of LNPs are semi-hexagonally close-packed. However, the stability of the structure might be disrupted after sectioning as evidenced by scanning transmission electron microscope (STEM) image of a 200 nm thick section from the LNPs photonic crystal, which showed that some particles had become detached, possibly due to weak interparticle interactions of LNPs (SI Fig. S4)."

Figure 3. (a) SEM image of the surface of photonic crystals.

Figure S4. STEM images of sectioned LNP-PC. The region of resin was highlighted by a yellow circle.

7. What are the potential applications of lignin photonic crystals?

Our response: Thank you very much for arising this question. As we understand from literature [1], lignin photonic crystals have potential applications in implanted/wearable optical devices, advanced cosmetics, and smart food packaging. We appreciate the broad importance of this question, and have highlighted the potential applications of lignin photonic crystals in the revised manuscript, as follows: “... potential applications in smart painting, rewritable paper, and invisible code.”

[1]. J. Wang, W. Chen, D. Yang, Z. Fang, W. Liu, T. Xiang, X. Qiu, Photonic Lignin with Tunable and Stimuli-Responsive Structural Color, ACS Nano. 16 (2022) 20705–20713. <https://doi.org/10.1021/acsnano.2c07756>.

8. What is the meaning of the particle sizes of lignin nanoparticles in Table 1? Hydrated particle sizes or dried particle sizes?

Our response: Thank you very much for the careful reading of our manuscript, and for raising this question. The particle sizes in Table 1 are theoretical sizes, i.e. calculated from a modified Bragg’s equation. They are neither hydrated particle sizes nor dried particle sizes. The implication of calculating size is to explain and guide which LNP particles size is responsible for which color.

9. Compared with the References named “Monodispersed Lignin Colloidal Spheres with Tailorable Sizes for Bio-Photonic Materials” and “Photonic Lignin with Tunable and Stimuli-Responsive Structural Color”, what are the advantages of this paper?

Our response: We agree that it is necessary to highlight the advantages of our paper and agree that our wording was not adequate. Our results make a significant contribution to understanding of the formation of lignin-based structural color from particle assemblies starting from *non-monodispersed* LNPs. We appreciate the reviewer’s question, and have highlighted advantages of this paper in the revised manuscript, as follows: “Here we show the possibility of forming lignin photonic crystals with rainbow coloration from non-monodispersed LNPs and improve the understanding of the central processes involved in the formation of the lignin photonic crystals.”

10. The structure colors of the prepared photonic crystals are not bright enough, and some necessary color characterization means are lacking;

Our response: Thank you very much for the careful reading of our manuscript, and for arising this question. We agree with the reviewer that the structure colors of the prepared photonic crystals are not bright enough. It is indeed a great challenge for lignin photonic crystals to maintain their bright color when changing from wet to dry state. We are planning to put more efforts into finding a solution for this challenge in the future. We appreciate the review's concern about the lack of necessary color characterization means, and have added new data in the supporting information, as follows:

Figure S5. CIE 1931 chromaticity coordinates of LNPs photonic crystal. Spectral power distribution of illuminant was D65.

11. The axes in Figure 5 are calibrated either inward or outward;

Our response: We really appreciate your suggestion. We have changed the axes in Figure 5 to be consistently outward, as follows:

12. Some of the pictures in this article are not clear enough.

Our response: Thank you very much for carefully reviewing our work. We appreciate the reviewer's concern about pictures in the article. We have improved or removed the pictures in the manuscript which are not clear enough. Please see the revised manuscript. We also note that the PDF compression during the submission compresses some of the images and reduces their display quality. We will provide high-resolution TIFF files of the figures prior to the final version of the article will be available.

Reviewer #3 (Remarks to the Author):

[Note from the editor: Please also see attached PDF]

Manuscript: Photonic crystals with rainbow colors by centrifugation-assisted assembly of colloidal lignin nanoparticles

Summary of content:

Liu et al. demonstrate a centrifugation-assisted assembly of lignin nanoparticles. Lignin is a bio-polymer which the authors synthesize into different nanoparticle sizes. After self-assembly, structural color is emitted from the structure covering the whole optical range. A study of the self-assembly dynamics is also provided.

Comments to the Manuscript:

Generally, the manuscript is well written apart from a few typos (see below) and due to its social relevance, I recommend the manuscript to be accepted after major revisions.

Our response: Thank you very much for your comments, for carefully reviewing our work, and for recommending it for publication after major revisions.

In the introduction, I am missing a comparison of advantages and disadvantages compared to cellulose based optics.

e.g. <https://onlinelibrary.wiley.com/doi/10.1002/adma.202109170>

Our response: We really appreciate the reviewer's suggestion that was addressed above, see our response to Reviewer 2.

I would recommend to remove the bracket from the following sentence in the introduction as it does not help the understanding.

Visible coloration results when the lattice constant (~~the lengths a , b , and c of the 2-three cell edges meeting at a vertex, and the angles α , β , and γ between those edges~~) of the crystal is comparable or smaller than the wavelength of light and the bandgap corresponds to a wavelength in the visible range (400–800 nm).

Our response: We appreciate the reviewer's suggestion. We have removed the bracket in the revised manuscript, as follows: "Visible coloration results when the lattice of the crystal is comparable or smaller than..."

On page 4 of the result section, the abbreviation of the unit v/v/min should be once written out.

Our response: Thank you very much for carefully reviewing our work. We have clarified the abbreviation of the unit v/v/min in the revised manuscript, as follows: "... by varying the water dilution rate (volume of water/volume of lignin solution/total dilution time, i.e., v/v/min) during the solvent shifting method..."

In the context of discussing the optics, a measured refractive index and absorption of a lignin thin film would be very valuable.

Our response: We appreciate the reviewer's suggestion. We have added the measured refractive index of lignin in the revised manuscript and the absorption of lignin in supporting information, as follows:

Figure 3. (b) Refractive index of lignin acetone solution.

Figure S6. Absorption spectra of lignin and LNPs. Lignin solution and LNPs dispersion were coated on cover glass, respectively. The spectrum of cover glass was recorded as a blank.

In a few subfigures, the legends are missing. These are Fig. 3b, Fig. 5a,b,d,f. In general, the authors should pay attention to the color blind people. There are two websites where color compatibility can easily be checked.

Color Scheme Designer: <http://89.161.193.92/colordesigner/>

Color Brewer: <https://colorbrewer2.org/#type=sequential&scheme=BuGn&n=3>

Our response: Thank you very much for the careful reading of our manuscript, and for providing the two websites for checking color compatibility. We have added the legends or corresponding text in the subfigures, please see the revised manuscript.

Figure 3 and 4 are to a certain extent doubled. I would recommend to discuss the optical response in a separate figure with the measurements of refractive index. The beam spot size of the UV-Vis reflection spectra should be given. The image of the rainbow film is shown three times but is not providing extra information. I would restrict the presentation to Fig. 5a. Fig. 3c does not add any information. Fig. 4 b should be larger to allow for better reading. The SEM images in 4f and g should either be repeated in higher quality or moved to the SI.

Our response: Thank you very much for carefully reviewing our work. We really appreciate the reviewer's suggestions. We have added the beam spot size of the UV-Vis reflection spectra in the revised manuscript, as follows: "The beam spot size was 1.5 mm."

We agree that some information are repeated and some optical response are missing. We have improved Figure 3 and 4 accordingly in the revised manuscript, as follows:

Figure 3. Structural and optical properties of lignin and LNPs photonic crystal. (a) SEM image of the surface of photonic crystals. (b) Refractive index of lignin acetone solution. (c) Reflectance spectra of the photonic crystals from three pieces of violet, blue and green.

Figure 4. Characterization of the structure of LNPs photonic crystal at different color layers. (a) Digital microscope image of cross-section of a photonic crystal. (b) Size distributions of LNPs in different color layers of the photonic crystal. More than 100 particles were measured in SEM images from (c), (d) and Supplementary Fig. 8. The peaks are at 150 nm, 160 nm, 170 nm, 180 nm, 200 nm, and 240 nm respectively. SEM images of the cross-section of photonic crystal in violet (c) and red (d) layer.

The phrase The yellow peak is unclear is not really scientific and should be better phrased.

Our response: We appreciate the reviewer’s concern about this phrase and agree that our wording was not adequate. We have rephrased the phrase, as follows: “No sharp peaks were observed in the yellow light range.”

The calculation of the PDI of LNPs should be presented in the SI. As the shapes of the curves are really rough, it would be valuable to know how the data was treated. Were there any fits applied? Was the PDI only extracted from the one sharp sub-peak at highest intensity etc? How many NPs have been analyzed.

Our response: Thank you very much for carefully reviewing our work. We really appreciate the reviewer’s suggestion and agree that it is necessary to provide the calculation detail. To calculate the PDI, we measured 100 particles for each color layer of the crystals, and calculated their average sizes and standard deviations, respectively. We have added the calculation of the PDI of LNPs in the revised supporting information, as follows:

Table S1. Calculation of the PDI of LNPs.

	Fig. 2e	Fig. 4d	Fig. S8d	Fig. S8c	Fig. S8b	Fig. S8a	Fig. 4c
Peak (nm)	/	240	200	180	170	160	150
Average size (D) (nm) *	77.4	226.8	193.0	190.7	175.5	156.0	154.2
Standard deviation (σ) (nm)	20.3	46.7	30.1	30.4	27.1	17.7	14.5
PDI=$(\sigma/D)^2$	0.069	0.042	0.024	0.020	0.030	0.013	0.009

* 100 particles were analysed.

Comment to Figure 5b: The r and n in supernatant are very close together and appear nearly as a m.

Our response: We appreciate the reviewer's comment. We have changed the text in Figure 5b, as follows:

Figure 5. (b) DLS size distribution of LNPs in pellet and supernatant after centrifugation at $3800 \times g$ for 2 h.

In the microscopy section in the characterization, it is mentioned that a STEM was used but there is no single STEM image neither in the manuscript nor in the SI.

Our response: We really appreciate the reviewer's comment. We have added STEM images in the revised supporting information (Figure S4), please see the response to Reviewer 2 Question 6.

Summary:

In summary, the content of the manuscript is relevant for a substantial part of Nature Communications target readership. The relevance of sustainable photonic components is very high. The presentation of scientific findings is consistent. Before publication, the discussion of the optical properties should be extended and the Figures 3 and 4 of the characterization should be better organized.

Our response: Thank you very much for reviewing our manuscript, highlighting its strengths and shortcomings. We also appreciate very much for the reviewer's careful reading and pointing out the inadequacy of the discussion of optical properties and the organization of Figures 3 and 4. We really appreciate the reviewer's suggestions and have discussed more about the optical properties in the revised manuscript, as follows: "We measured the refractive index of lignin by dissolving different amount of lignin in acetone (Fig. 3c). The refractive index of lignin was calculated to be 1.55 by extrapolating from a fitting line ($y = 0.0019x + 1.3594$, $R^2 = 0.96$) and found to be similar as previously reported value⁴³. " and "In addition, both lignin and LNPs did not show the ability of absorbing only specific wavelengths of visible light (SI Fig. S6), which indicated that they are different from light absorbing pigments."

We have reorganized the Figures 3 and 4, please see the revised manuscript.

Typos and small remarks:

Abstract:

-In situ optical microscope images provided additional evidence on the importance of dynamic rearrangement of LNPs during their **assembly** into semi-closed packed structures.

Introduction:

-The brown-black color palette of industrial lignin limits the opportunity window for **optical** applications.

Results:

-To answer this question, we can look into **the** origin of this phenomenon.

-The size-dependent structural **a** color of the photonic crystals was predicted by theoretically calculated particle diameters of LNPs (Table 1).

-To image samples with SEM, we sputter coated the surfaces of the samples with gold, **which did not change their color as observed color appearance** (SI Fig. S3)

Comment: Please rephrase the yellow indicated part, to make the meaning clearer.

-RCF explain abbreviation at the first occasion at which it occurs.

-DLVO abbreviation should be written out at the first occasion at which it occurs.

Methods:

-Lignin was first purified with ethanol, and then was extracted **ed** with acetone, according to a method reported in the literature.

-LNPs **were** prepared by similar solvent shifting method from literature.

-The fractionated lignin solution (solid content: ~ 5 g) was added to 100 g DI water and **stirred** under 600 rpm for 1 h.

Our response: Thank you very much for careful reading of our manuscript and for pointing out the unclear statements, missing explanations of abbreviations, and spelling mistakes. We have carefully proofread the manuscript. We have revised and corrected a new version of the manuscript accordingly. Please see the revised manuscript. We have rephrased some sentences, as follows:

-“To image samples with SEM, we sputter coated the surfaces of the samples with gold. No change in color was observed after coating compared to the digital microscope image before coating (SI Fig. S3).”

-“Relative centrifugal force (RCF) can be converted from revolutions per minute (rpm), using the following formula: $RCF = (rpm)^2 \times 1.118 \times 10^{-5} \times r$ (r: centrifugation radius in cm).”

-“The DLVO (Derjaguin, Landau, Verwey, and Overbeek) theory...”

REVIEWERS' COMMENTS

Reviewer #1 (Remarks to the Author):

The concerns has been answered, nut the format of the newly added reference is different from the other ones.

Reviewer #2 (Remarks to the Author):

Some of the answers are more detailed, but some of the questions remain unclear:

1. Innovative or not prominent, I suggest writing more fully;
- 2.The advantages of the method should be described in more detail ;
- 3.“The thickness of the LNPs photonic crystal was 1.08 mm”, Please verify the accuracy ;
- 4.What is the stability of the structure of lignin photonic crystals ? Do photonic crystal structures fall off when they are subjected to friction, washing and rubbing?

Reviewer #3 (Remarks to the Author):

Dear authors,

I appreciate the work you have put into revising the manuscript. All comments have been addressed. I think the manuscript has gained significantly from the adjustments and I recommend it for publication.

Responses to Reviewer's comments:

We would like to thank the expert reviewers again for their comments on our revised manuscript. Our point-by-point responses are included in blue font after each reviewer comments that are available in black. The changes made to the manuscript are shown in red color.

Reviewer #1 (Remarks to the Author):

The concerns has been answered, nut the format of the newly added reference is different from the other ones.

Our response: Thank you very much for reviewing our manuscript. The format of the newly added reference has been updated to remain consistency with the other references.

Reviewer #2 (Remarks to the Author):

Some of the answers are more detailed, but some of the questions remain unclear:

1. Innovative or not prominent, I suggest writing more fully;

Our response: Thank you very much for reviewing our manuscript. We have highlighted more our innovation in the abstract, as follows: "Our results show that centrifugation is crucial for the formation of lignin photonic crystals, because assembly of lignin nanoparticles without centrifugation assistance leads to the formation of strip patterns rather than photonic crystals. We further prove that the functions of centrifugation are to classify lignin nanoparticles according to their particle size and produce monodispersed particle layers that display gradient colors from red to violet."

2.The advantages of the method should be described in more detail ;

Our response: We have rephrased the statement, please see the response to the Question 1 above along with the revised abstract.

3."The thickness of the LNPs photonic crystal was 1.08 mm", Please verify the accuracy ;

Our response: We have modified the statement, as follows: "The thickness of the LNPs photonic crystal was 1.1 mm (accuracy: ± 0.1 mm)"

4.What is the stability of the structure of lignin photonic crystals ? Do photonic crystal structures fall off when they are subjected to friction, washing and rubbing?

Our response: It would be interesting to further study the stability of the structure of lignin photonic crystals. We have added the statement in the revised manuscript, as follows: "Future studies can investigate the stability of lignin photonic crystal structures in processing applications, particularly their resistance to mechanical and solvent stresses."

Reviewer #3 (Remarks to the Author):

Dear authors,

I appreciate the work you have put into revising the manuscript. All comments have been addressed. I think the manuscript has gained significantly from the adjustments and I recommend it for publication.

Our response: Thank you very much for reviewing our manuscript.